# Complex picture for likelihood of ENSO-driven flood hazard

R. Emerton[1,2,3], H.L. Cloke[1,2], E.M. Stephens[1], E. Zsoter[1,3], S.J. Woolnough[4] & F. Pappenberger[3]

El Niño and La Niña events, the extremes of ENSO climate variability, influence river flow and flooding at the global scale. Estimates of the historical probability of extreme (high or low) precipitation are used to provide vital information on the likelihood of adverse impacts during extreme ENSO events. However, the nonlinearity between precipitation and flood magnitude motivates the need for estimation of historical probabilities using analysis of hydrological data sets. Here, this analysis is undertaken using the ERA-20CM-R river flow reconstruction for the twentieth century. Our results show that the likelihood of increased or decreased flood hazard during ENSO events is much more complex than is often perceived and reported; probabilities vary greatly across the globe, with large uncertainties inherent in the data and clear differences when comparing the hydrological analysis to precipitation.

[1] Department of Geography and Environmental Science, University of Reading, Reading RG6 6AB, UK. [2] Department of Meteorology, University of Reading, Reading RG6 6BB, UK. [3] European Centre for Medium-Range Weather Forecasts, Reading RG2 9AX, UK. [4] National Centre for Atmospheric Science, Department of Meteorology, University of Reading, Reading RG6 6BB, UK. Correspondence and requests for materials should be addressed to R.E. (email: r.e.emerton@pgr.reading.ac.uk).

El Niño Southern Oscillation (ENSO) is the most prominent pattern of inter-annual climate variability[1], and is known to influence river flow[2] and flooding[3–5] at the global scale. In the absence of hydrological analyses, products indicating the likelihood of extreme precipitation are often used as an early indicator of flooding during extreme ENSO events[6]. However, the nonlinearity between precipitation and flood magnitude and frequency[7] means that it is important to assess the impact of ENSO not just on precipitation, but on river flow and flooding. This is especially important as, as stated by Chiew and McMahon[2], 'it is likely that the streamflow-ENSO relationship is stronger than the rainfall-ENSO relationship because the variability in rainfall is enhanced in runoff and because streamflow integrates information spatially'.

Here, a global scale hydrological analysis is performed to estimate the historical probability of increased or decreased flood hazard in any given month during El Niño/La Niña events, assessing the added benefit of directly analysing river flow over the use of precipitation as a proxy for flood hazard.

Historical probabilities provide useful information about typical ENSO impacts based on historical evidence[8,9] and are, as stated by Mason and Goddard[8], 'a better estimate of the future climate than the assumption that seasonal conditions will be the same as average'. Nonetheless, there are some key considerations when using such information. One such consideration is that no two El Niño events are the same[8,10]; differences in the peak amplitude, temporal evolution and spatial pattern of warming are likely to affect the timing and magnitude of the resulting impact on river flow. There are many suggested ways to classify ENSO diversity[11], for example, El Niño events are often described as 'East Pacific' (EP) or 'Central Pacific' (CP), dependent on where the peak warming occurs. While this is an over-simplification of the complexity surrounding ENSO diversity, the location of the peak warming can alter the influence on river flow. An additional consideration is the influence of warming ocean temperatures on ENSO events and their related impacts. Recent studies[12,13] suggest that projected changes in the Walker circulation and associated weakening of equatorial Pacific ocean currents are expected to result in more frequent, and more extreme, El Niño and La Niña events[12,14].

In the past, studies have been limited to reanalysis data sets of no longer than ∼40 years[3–5], in which there is a sample of ≤10 El Niño and ≤13 La Niña events, or observational data with inconsistent coverage, both spatially and temporally[2]. We have created a twentieth century (1901–2010) model reconstruction of river flow in order to obtain a hydrological data set with consistent global coverage over an extended time period. Research by Essou et al.[15] indicates that global meteorological reanalysis data sets 'have good potential to be used as proxies to observations' in order to force hydrological models, particularly in regions where few observations are available. This data set was created by forcing a research version (described in the Methods) of the Global Flood Awareness System[16,17] (GloFAS) with the ERA-20CM[18] meteorological model reconstruction of the European Centre for Medium-Range Weather Forecasts (ECMWF) to produce a 10-member, 0.5° resolution reconstruction of river flow (from here on, ERA-20CM-R) containing 259,200 grid points covering the global river network (Supplementary Fig. 1). Figure 1 depicts a time series of three key variables used in this study, alongside the timing of the 30 El Niño and 33 La Niña events identified in ERA-20CM-R (see Methods).

Previous work by Ward et al.[4] has looked at the influence of El Niño on flood return periods, quantifying the percentage anomaly during El Niño years in comparison with climatology (defined as the long-term average of historical river conditions or meteorological parameters). To ensure accurate estimation of historical probabilities of ENSO-driven flood hazard, this analysis was replicated using the new ERA-20CM-R data set and gives similar results (Supplementary Fig. 2).

In this study, using a climatology of all years and all El Niño/La Niña years, we calculate the percentage of past El Niño/La Niña events during which the river flow fell in the upper (lower) quartile of climatology, defined here as 'abnormally high (low) flow'. Our results show that the likelihood of increased or decreased flood hazard during ENSO events is much more complex than is often perceived and reported; probabilities vary greatly across the globe, with large uncertainties inherent in the data and clear differences when comparing the hydrological analysis to precipitation.

## Results

**Historical probabilities during El Niño.** Figure 2a shows the historical probabilities for February during an El Niño, with the full set of El Niño and La Niña results presented in Supplementary Figs 7 and 8, respectively. El Niño events tend to span two calendar years, evolving in boreal spring and reaching their peak magnitude in winter of the same year, before decaying into the following spring/summer. Shortly after the peak, February sees some of the highest probabilities and extensive spatial coverage of regions influenced by El Niño (where >40% probability of abnormally high or low river flow represents a significant influence); 34.5% of the land surface indicates a significant increase in the probability of abnormally high or low river flow (19.2% for high, 15.3% for low) compared to any given year.

The influence of El Niño on river flow can be seen as early as June (see Supplementary Fig. 7), shortly after ENSO tends to move into the warm phase, with some regions, mostly confined to the tropics, beginning to see up to a 50% probability of high or low river flow in the ensemble mean. In August and September, much of South America, south of the Amazon River, is somewhat likely (∼40–60% probability) to observe higher than normal river flow; however, in November, closer to the typical peak of El Niño events, a reversal to drier conditions across much of Brazil is observed. The southern USA has a high probability (up to 70%) of high river flow from December onwards, while Mexico is another region that experiences a reversal in the influence of El Niño, from decreased flood hazard up until September/October, to increased flood hazard from November onwards. Other regions are much more consistent, such as Indonesia, which has a high certainty of abnormally low river flow throughout the evolution, peak and decay of El Niño. However, it is important to note that across the globe, the uncertainty around these probabilities can be high.

**Evaluating the uncertainty.** Indeed, the historical probabilities themselves give an indication of the uncertainty in the response of the river flow to ENSO events. Here, the 10 ensemble members of ERA-20CM-R also allow interpretation of the uncertainty in the data set, as each ensemble member represents an equally probable reconstruction of the river flow. To provide an indication of this uncertainty, Fig. 2b shows the range of the probability around the mean probability shown in Fig. 2a. The influence of El Niño is much more certain in some locations; for example, in coastal Ecuador/northern Peru, the probabilities vary by only 9%. These locations (darkest shading, 5–10% range) stand out in Fig. 2b; these are the areas where there is potential to use such historical probabilities as an early indicator of increased or decreased flood hazard, as they tend to give high probabilities combined with small uncertainties. However, much of the globe shows a range of

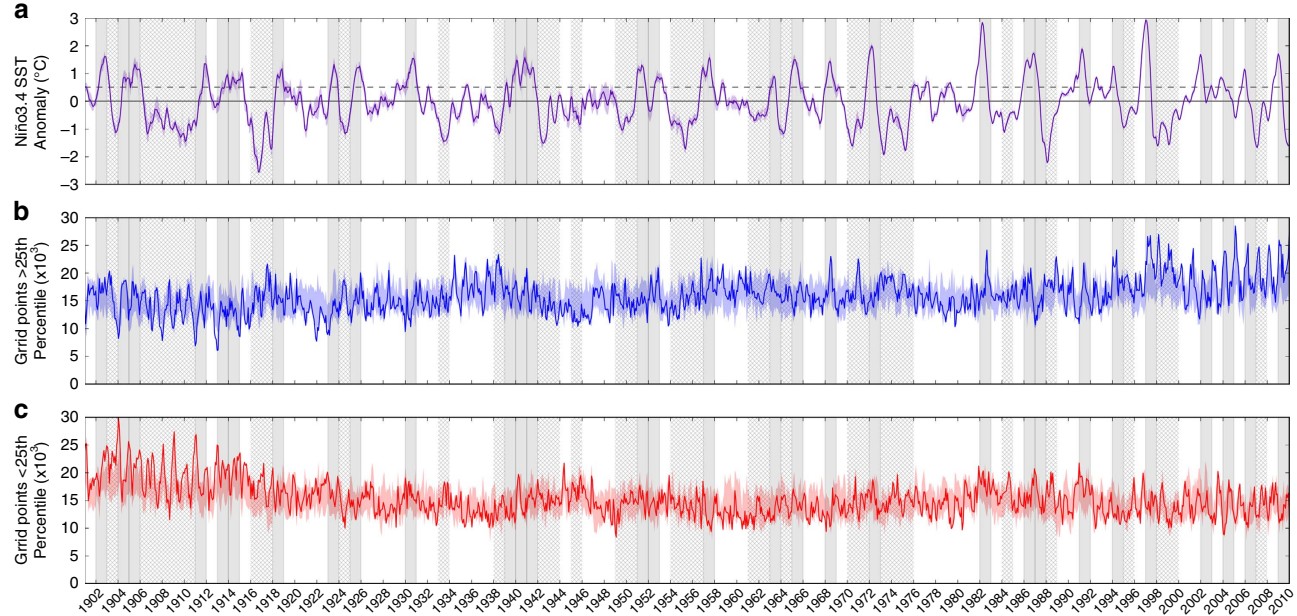

**Figure 1 | Time series of three key ERA-20CM-R variables and timing of El Niño and La Niña events.** (**a**) Three-month running mean sea surface temperature anomaly in the Niño3.4 region (SSTA3.4), and number of grid points globally in which monthly mean river flow (**b**) exceeds the top 25th percentile and (**c**) falls below the lower 25th percentile. Solid lines show the mean of the 10 ensemble members, while shading indicates the spread of the members. The SSTA3.4 is used to identify El Niño and La Niña years in the data set, highlighted here by the grey shaded and hatched bars, respectively.

20–40%, and some small regions, such as in northwest Spain and eastern Argentina, see a range up to 70% across the ensemble members. The implication is that while some regions see high probabilities of increased flood hazard (e.g., up to 77% in northern Peru), across much of the globe the likelihood is much lower and much more uncertain than might be useful for decision-making purposes.

**Importance of the hydrology**. Evaluating the historical probabilities of abnormally high or low precipitation, using the ERA-20CM precipitation data set, confirms that there is additional information which can be gained from the hydrological analysis. For example, parts of northern Africa are likely to see high precipitation in February (Supplementary Fig. 3a); however, the River Nile is likely to see dry river conditions (Fig. 2a), indicating that the river is influenced more by upstream rather than local precipitation.

To further highlight the importance of considering the hydrological impacts, Fig. 3 indicates regions, shown in pink (green), where the probability of high river flow is greater (smaller) than that of high precipitation. These differences suggest that the influence of El Niño is more pronounced in the river flow in pink regions, and conversely, green highlights regions where the use of precipitation as a proxy for flood hazard results in an overestimation of the probabilities. This could also indicate that the region is likely to experience a lagged influence of El Niño on river flow. The corresponding results for low flow are presented in Supplementary Fig. 4.

**Historical probabilities during La Niña**. El Niño events are often followed by a La Niña, the cool phase of ENSO. While La Niña events tend to be less widely discussed in the media, their influence on precipitation is often used as a proxy for flood hazard, as with El Niño. We have therefore extended this analysis to evaluate the probability of increased (or decreased) flood hazard during La Niña years. We find that many regions

influenced by El Niño are likely to observe the opposite response during La Niña. Figure 4 shows these probabilities, again for February, during a La Niña event, with the full set of results shown in Supplementary Fig. 8. It is evident that less of the land surface is significantly influenced by La Niña compared to El Niño during this month (22% of the land surface compared to 34.5%). Probabilities, while still significant, also tend to be lower than for the same month during an El Niño; the highest probability of increased flood hazard shown in Fig. 4a is 67, and 69% for decreased flood hazard. Again, the uncertainty surrounding this mean probability is large (20–40% and in some areas >70%) across much of the globe; this can be seen in Fig. 4b.

**Maximum probabilities during El Niño/La Niña**. While the monthly maps of historical probabilities give an indicator of the probability of increased (or decreased) flood hazard and when this is likely to occur, it is perhaps useful to consider the event as a whole, as the peak conditions occur at different times across the globe. Figure 5a (b) shows the maximum probability of increased flood hazard during any month of an El Niño (La Niña) event; this provides an overview of whether a region is likely to experience a change in river conditions or not during or following the event. Figure 5 also indicates where the uncertainty surrounding the probabilities is high; this tends to be where the probability is lower, while regions with high probabilities also indicate higher certainty. This analysis further confirms that across much of the globe, such historical probabilities are much more uncertain than is often communicated. The corresponding results for decreased flood hazard are shown in Supplementary Fig. 5.

**Comparison with observations**. A comparison of the historical probabilities against observed data sets was also undertaken (see Methods and Supplementary Fig. 6). While this proved challenging at the global scale due to a lack of consistent and extensive river flow records in regions of the world where ENSO events

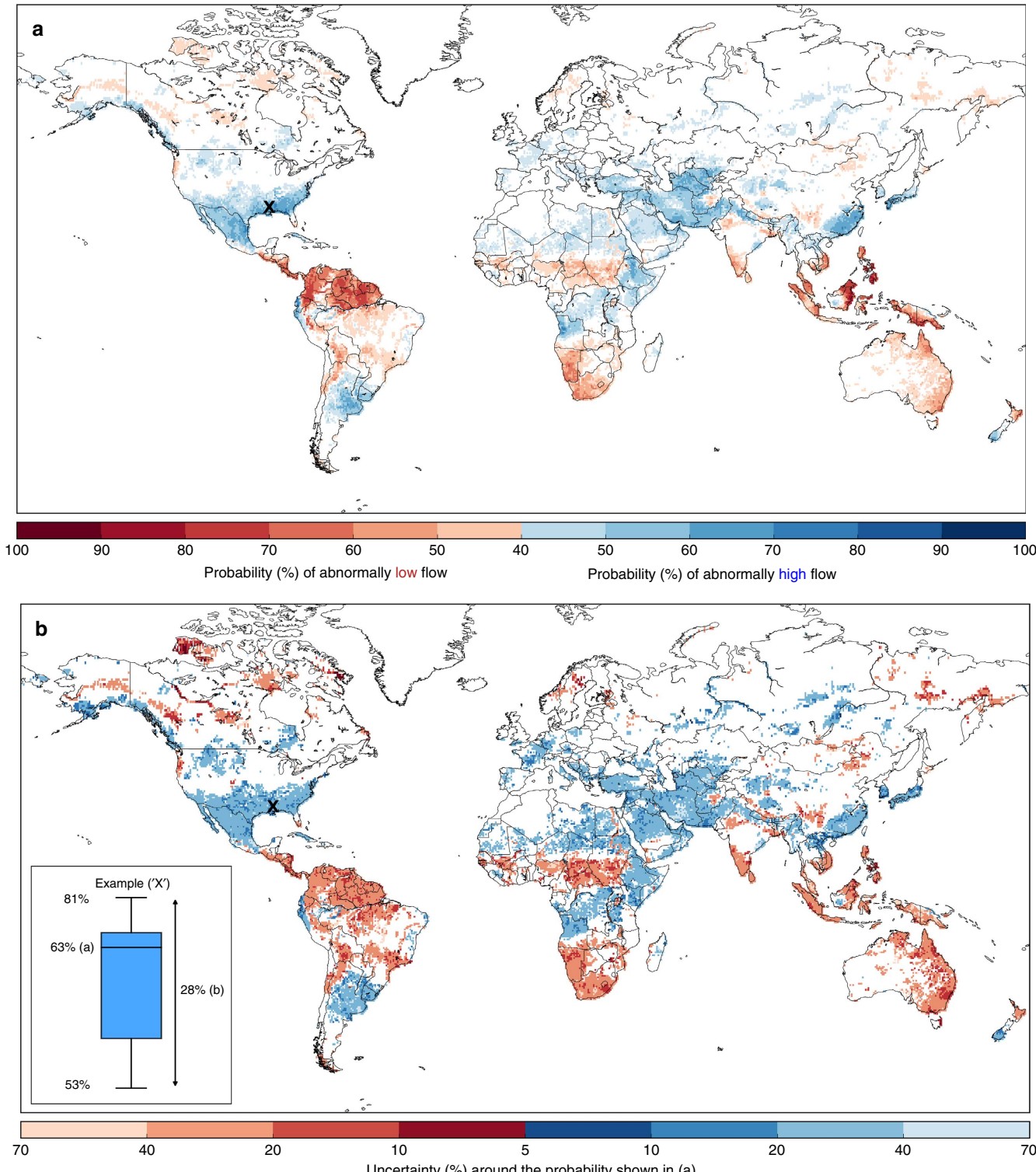

**Figure 2 | Historical probability of increased or decreased flood hazard during one month of an El Niño.** (**a**) Probability of abnormally high (blue) or low (red) monthly mean river discharge. Based on the mean of the 10 ERA-20CM-R ensemble members exceeding the 75th percentile, or falling below the 25th percentile, of the 110-year river discharge climatology. (**b**) Uncertainty around the probability shown in (a), i.e., the difference between the minimum and maximum of the 10 ensemble members (%). The boxplot (**b**, inset) gives an example graphical representation of the uncertainty range at one grid point, marked on the map by an 'x', where the mean probability indicated in (a) is 63%. The range is given by the difference between the minimum and maximum of the 10 ensemble members; in this case 53 and 81%, giving a 28% range falling in the 20–40% bracket in (b). The month of February is chosen as, occurring shortly after the peak of an El Niño, it sees extensive spatial coverage of land areas influenced by El Niño.

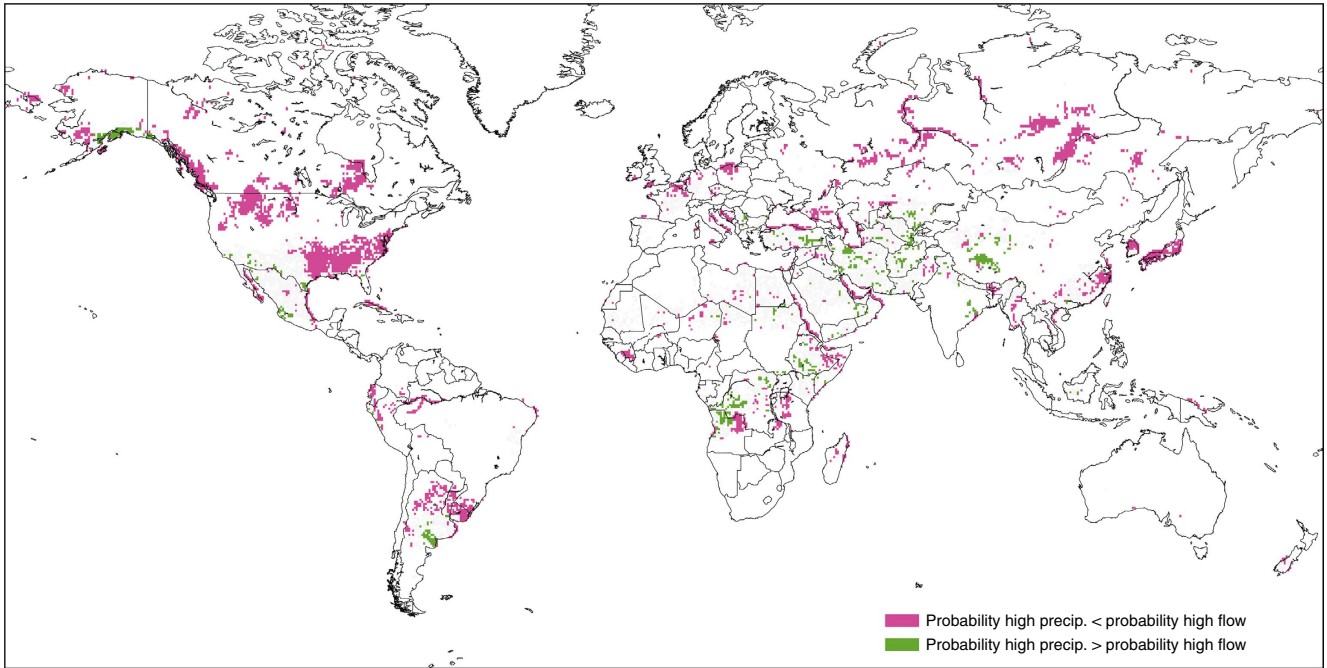

**Figure 3 | Comparison of historical probabilities based on precipitation and river flow.** Regions where the difference in probability of abnormally high precipitation compared to probability of high river flow, in the month of February during an El Niño, is greater than 10% (based on the ensemble mean). Pink shading indicates that the probability of high precipitation is smaller than the probability of high river flow, while green shading indicates that probabilities are larger for precipitation.

have the most influence, the evaluation suggests a potential overestimation of the probabilities in both the precipitation and river flow reconstructions. This stresses that while these model reconstructions are currently the best available data for such research, there is a need for more extensive river flow observations in regions impacted by ENSO events.

Throughout the results, the complexity and uncertainty surrounding such historical probabilities is evident. Indeed, observations of flooding in February 2016, during the strong 2015–2016 El Niño event, reflect this complex picture of ENSO-driven flood hazard. The expected flooding (based on the results shown in Fig. 2a) in Peru, Bolivia, Argentina and Angola was observed[19]; yet in several other regions, such as Eastern China, Japan and parts of the Middle East, no flood events were recorded. Flooding also occurred in Indonesia despite a high likelihood of dry river conditions. In Kenya and Peru, two examples where flood preparedness actions were taken ahead of El Niño, flooding was much less severe than expected[20,21]. A recent Nature correspondence[22] also highlighted the unexpected winter weather in the USA; California experienced heatwaves rather than prolonged rain events, while Seattle was expecting a worsening drought and instead endured the wettest winter on record (see also Supplementary Fig. 7).

## Discussion

We have conducted a global hydrological analysis of ENSO as a predictor of flood hazard based on historical probability estimates using a new, extended-length model reconstruction of river flow. The importance of addressing the hydrology in addition to precipitation is evident in the differences between the probabilities of high river flow and precipitation, and in the ability to further evaluate areas likely to see a lagged influence of El Niño on river flow. We conclude that while it may seem possible to use historical probabilities to evaluate regions across the globe that are more likely to be at risk of flooding during an El Niño/La

Niña, and indeed circle large areas of the globe under one banner of wetter or drier, the reality is much more complex. It is therefore important to undertake research that focuses on the region(s) of interest and consider the impact of ENSO diversity and other drivers of climate variability on the hydrology and flood hazard.

## Methods

**The new twentieth century river flow data set.** For this study, we have created a twentieth century (1901–2010) reconstruction of river discharge, in order to obtain a data set with consistent global coverage over an extended time period. This was achieved by forcing an alternative setup of the GloFAS[16,17] with the 10 ensemble members of the ERA-20CM[18] atmospheric model ensemble of the ECMWF to produce a 10-member ensemble of river discharge for the global river network (ERA-20CM-R).

The operational set-up of GloFAS takes the runoff output from the ECMWF Integrated Forecast System (IFS) and runs this through the Lisflood hydrological routing model[16]. Here, we instead use the Catchment-based Macro-scale Floodplain[23] (CaMa-Flood) routing model to create the river discharge reconstruction at 0.5° resolution from the gridded ERA-20CM runoff data. A map of the CaMa-Flood global river network is given in Supplementary Fig. 1. We note here that the version of GloFAS used in this study is uncalibrated.

While the use of the ERA-20CM model reconstruction allows a consistent analysis at the global scale, and provides a much longer time period over which to study these extreme events, there are limitations that must be considered. ERA-20CM incorporates ENSO and twentieth century climate trends, and assimilates sea-surface temperature and sea ice cover[18]. It does not, however, assimilate atmospheric observations. This is a drawback as the model reconstruction is able to provide a statistical estimate of the climate, but is not able to reproduce synoptic situations. We have therefore undertaken a comparison with the best available precipitation and river discharge observations for the twentieth century and are satisfied that ENSO teleconnections are well-represented in ERA-20CM(-R). Of course, there is further uncertainty introduced when going back as far as the early twentieth century when fewer observations were available; the 10 ensemble members go some way to representing this uncertainty and are a key benefit of this particular data set, and thus are considered throughout this study.

**Identifying the El Niño years.** To conduct this analysis, we first identified the El Niño/La Niña years in the data set. This was done using the definition that the US National Oceanic and Atmospheric Administration (NOAA) use to declare El Niño (La Niña) conditions operationally[24]. This definition states that the sea

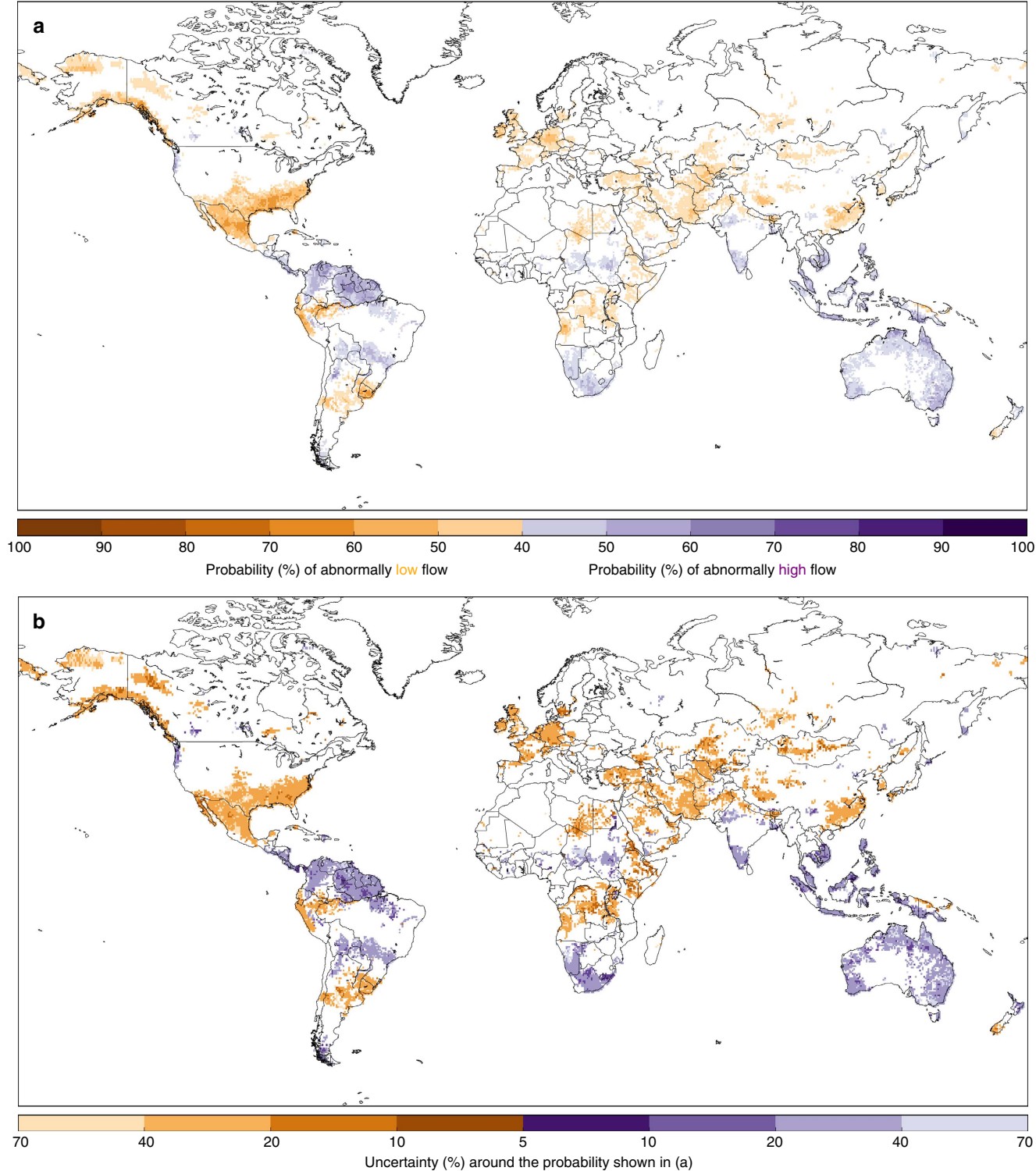

**Figure 4 | Historical probability of increased or decreased flood hazard during one month of a La Niña.** (**a**) Probability of abnormally high (purple) or low (orange) monthly mean river discharge in the month of February during a La Niña. Based on the mean of the 10 ERA-20CM-R ensemble members exceeding the 75th percentile, or falling below the 25th percentile, of the 110-year river discharge climatology. (**b**) Uncertainty around the probability shown in (a), i.e., the difference between the maximum and minimum of the 10 ensemble members (%).

surface temperature (SST) anomaly must remain $\geq 0.5\,^{\circ}\mathrm{C}$ ($\leq 0.5\,^{\circ}\mathrm{C}$), in the Niño3.4 region in the central Pacific (5° S–5° N, 170°–120° W), for at least five consecutive 3-month periods. Here, we extracted the ERA-20CM SST data and calculated the 3-month running mean SST anomalies for the Niño3.4 region, allowing identification of the 30 (33) years in which El Niño (La Niña) conditions were present from 1901 to 2010. These are listed in Supplementary Table 1, where the El Niño/La Niña year refers to the year in which the event evolves and typically

also reaches its peak, as ENSO events often span 2 years, decaying into the following year. We note that while there is generally a good agreement between the ENSO events identified in ERA-20CM and those published by NOAA[25] for the same period, there are, however, some discrepancies. This is likely due to the different indices/definitions used to identify the ENSO events. For example, in 1977 and 1979, El Niño events are identified by NOAA, using the Multivariate ENSO Index[25], but these are not picked up in this study. In Fig. 1, it is evident that the

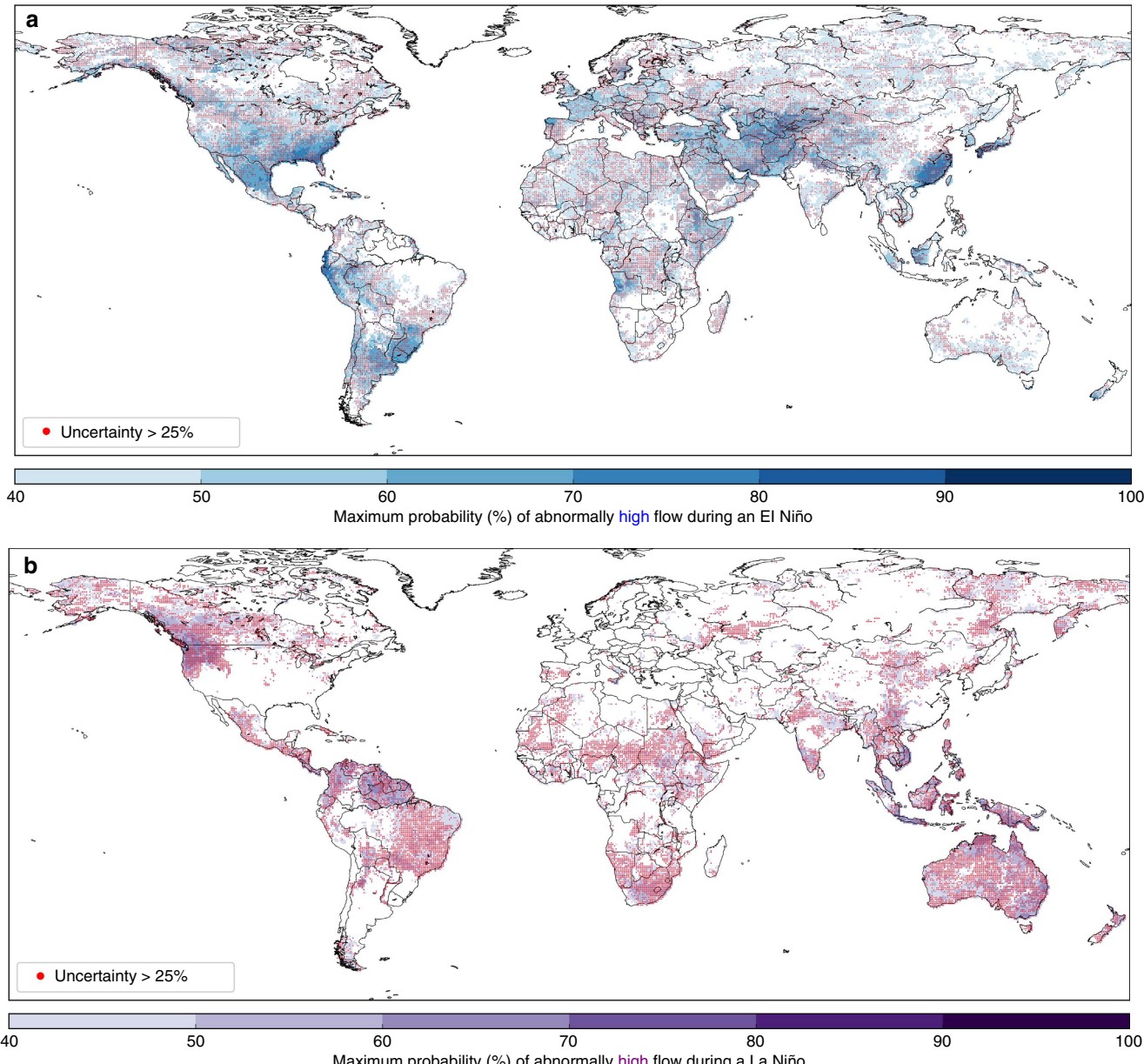

**Figure 5 | Maximum probability of increased flood hazard during an ENSO event.** Maximum probability of abnormally high river flow in any month during (**a**) an El Niño event and (**b**) a La Niña event. Based on the mean of the 10 ERA-20CM-R ensemble members exceeding the 75th percentile, or falling below the 25th percentile, of the 110-year river discharge climatology during, or shortly after the decay of, an ENSO event. Stippling indicates where the uncertainty surrounding this probability is high, i.e., the range of the ensemble members exceeds 25% probability.

SST did exceed 0.5 °C in ERA-20CM, but this did not persist for long enough to be identified as an event. This is a limitation of the need to use one of the many varying methods of classifying and identifying ENSO events. This method was chosen as it is the most operationally relevant at the time of writing.

**Historical probability estimation.** For the results presented in this study, the 110-year ERA-20CM-R climatology was used to calculate the upper and lower 25th, 10th and 5th percentiles of river discharge for every grid box. The historical probability of abnormally high or low river flow in any given month was then estimated, through calculation of the percentage of the 30 (33) identified El Niño (La Niña) years in which the river discharge exceeded (high flow) or fell below (low flow) the three percentile thresholds, for each of the 10 ensemble members of ERA-20CM-R. The analysis presented in this paper is based on percentiles so as to avoid potential large errors caused by bias in the data set compared to observations (discussed further below).

Maps of the resulting probabilities were produced based on the mean of the 10 ensemble members. As the number of ENSO events cover a substantial part of the 110-year period, there is a chance of picking up random effects. The maps produced therefore only display results where the probability is significantly greater

than normal, i.e., $\geq 40\%$; an 'event' (occurrence of abnormally high or low flow) with a probability of 40% during one month of an El Niño/La Niña has only a 5% chance of occurring by chance in that month, and thus represents a significant increase in the probability compared to the likelihood of occurring at random.

Additionally, the spread in the ensemble members is designed to reflect the uncertainty in the data set, and can indicate a range of possible outcomes or probabilities. As such, we have further calculated the uncertainty around the mean probability for the whole globe, based on the range across the ensemble members. For each ensemble member, the range between the minimum and maximum ensemble members was calculated for every grid box individually. This allows us to interpret the uncertainty in the probability caused by uncertainty in the data set.

El Niño/La Niña onset tends to occur in boreal spring/early summer and peak in winter[25], before decaying into the following spring. As such, the monthly analysis was undertaken for a period of 2 years; the year of onset, and the following year during which the El Niño/La Niña decays, in order to capture any lagged influence on river flow. Significant influence is shown in the results from June during the El Niño/La Niña year, to the following September (16 months). While it would seem advantageous to summarize the findings by season for simplicity, evaluation of the results shows that the patterns of influence across the globe can change dramatically, in some instances, from one month to the next. Summarizing

these maps into seasons may therefore result in a loss of information for some months.

**Difference between river flow and precipitation.** A key aim of this paper was to evaluate the added benefit of the hydrological analysis over the use of precipitation as a proxy for flood hazard. To do this, the same method used to estimate the historical probabilities in the river flow reconstruction (ERA-20CM-R) was also applied to the ERA-20CM precipitation reconstruction. The horizontal resolution of the ERA-20CM precipitation data is ∼125 km, while the river flow data is at 0.5° (∼55 km) resolution. To compare these, the results from the precipitation data were remapped to the higher resolution of the river flow data using a simple nearest neighbor remapping algorithm. The difference between the historical precipitation probabilities and river flow probabilities was then calculated for the mean of the 10 ensemble members.

**Comparison with observations—precipitation.** To evaluate the results shown using the new ERA-20CM(-R) data set, the same method for estimating historical probabilities was also applied to other, related data sets: the Global Precipitation Climatology Centre (GPCC) Full Data reanalysis (GPCC-FD)[26] at 0.5° resolution, and the Global Runoff Data Centre (GRDC) river discharge observations[27]. Again, percentiles are used throughout to allow reliable comparison with observations despite potentially large bias in the model reconstruction values compared to observed values.

The GPCC-FD reanalysis is a global gridded precipitation data set based on interpolated rain gauge data[26]. Comparing the ERA-20CM and GPCC-FD precipitation data sets indicates that the regions influenced by El Niño are well-represented by ERA-20CM (see Supplementary Fig. 3b), and in line with well-known ENSO-sensitive regions, such as Australia, Indonesia, Argentina (the Rio de la Plata delta) and the southern USA—which have been shown to be well-represented in the GPCC-FD[28]. However, the strength of this link appears to be overestimated compared to observations, as the ERA-20CM data show higher probabilities of abnormally high or low precipitation than the GPCC-FD. Some of this overestimation may be caused by the use of the ensemble mean to produce the ERA-20CM maps, as averaging across the 10 ensemble members likely results in a reduction of the variance and we therefore pick up the forced part of the signal.

**Comparison with observations—river discharge.** As no gridded observational data set of river discharge exists for the global river network, archived station data from the GRDC were used. Criteria for data suitability were chosen to identify those stations which could be of use in this study. Firstly, only stations with at least a 75-year record of observations between 1901 and 2010 were included; these could be stations recording on a daily or monthly basis. Of these, any stations with more than 50% of the data missing were removed. In total, 1287 stations fit the criteria (232 monthly, 1,055 daily), of which the majority have <30% of the data missing. Each of these stations were manually checked to ensure that they correspond to the correct river point (taking into account location and upstream area) on the model river network. A key limitation of using the GRDC observations for this study is that many of these stations lie in river basins outside of the tropics and subtropics—the regions that tend to be most strongly influenced by ENSO events. This highlights the need for more consistent global river flow observations, but in their absence, model reconstructions and reanalyses present the best available data for regional and global scale research based on historical evidence.

To compare the results based on observations with ERA-20CM-R, we produced a reliability diagram (Supplementary Fig. 5) for the historical probability of abnormally high river flow, comparing the forecast (historical) probability of an event (in this case, river flow exceeding a given percentile) with the observed frequency of the event. This was achieved by first locating all grid points in the ERA-20CM-R data set that contain a GRDC station that fit the criteria outlined above. For each percentage band (in 10% bins, as displayed on the maps shown in the Results) of the 'forecast', the observed frequency of river flow exceeding the upper 25th, 10th and 5th percentiles of the 110-year climatology was calculated for each GRDC station, before taking the mean across all stations, and all 16 months used in the analysis (June to the following September). This allows comparison of the predicted probability with the observed frequency. The reliability diagram (Supplementary Fig. 5) and the discrepancy between forecasted and realized probabilities indicates that there is a potential overestimation of the forecasted probabilities. There are limitations, however, in that we have very few, or no, observation stations with which to compare the results for the higher probabilities (Supplementary Fig. 5, inset), particularly in regions that are most significantly influenced by El Niño/La Niña and where reliability may be better, such as the tropics. This suggests that such a reliability analysis may not be fully representative of the results. Additionally, the data records vary from station to station; therefore, the number of El Niño/La Niña years included in the observational record of each station also varies.

**Data availability.** The ERA-20CM, GPCC-FD and GRDC data that support the findings of this study are publicly available online at http://www.ecmwf.int/en/research/climate-reanalysis/era-20cm-model-integrations, http://www.dwd.de/EN/ourservices/gpcc/gpcc.html and www.bafg.de/GRDC. The ERA-20CM-R data that support the findings of this study are available from the corresponding author upon reasonable request.

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

## Acknowledgements

This work has been funded by the Natural Environment Research Council (NERC) as part of the SCENARIO Doctoral Training Partnership under grant NE/L002566/1. E.Z. was supported by the Copernicus Emergency Management Service—Early Warning

Systems (CEMS-EWS (EFAS)). The time of E.M.S. was funded by Leverhulme Early Career Fellowship ECF-2013-492. H.L.C. and F.P. acknowledge financial support from the Horizon 2020 IMPREX project (grant agreement no. 641811, www.imprex.eu). S.J.W. was supported by the National Centre for Atmospheric Science, a NERC Collaborative Centre, under contract R8-H12-83.

## Author contributions

E.Z. produced the ERA-20CM-R data set; R.E. conceived and posed the research question, carried out the analysis and wrote the paper; R.E., H.L.C., E.M.S., F.P. and S.J.W. designed the study and interpreted the results. All authors commented on the manuscript.

## Additional information

**Competing interests:** The authors declare no competing financial interests.

