## [Peer Review File · Nature Communications]

Reviewers' comments:

Reviewer #2 (Remarks to the Author):

Remarks to the Author:

I have previously reviewed this paper when it was submitted to [redacted]. It has been redirected to Nature Communications with the option of having the same reviewers assigned. I have not kept a copy of the original submission but as far as I can tell, the paper remains essentially the same. As such, this review is largely based on the original one made earlier this year. I have gone through the paper once again and have updated my review.

This paper is a global assessment of the impact of ENSO flooding hazard over the 20th century. The most important result of this paper is that it recognizes that the role of ENSO in flooding is a lot more complex than often reported in the scientific literature, and especially by the media in general. Some of our very recent work over North-America has shown that the role of ENSO on precipitation and discharge has generally been overstated and that other manifestations of natural variability have much more control than ENSO on hydrological conditions. For example, the 2001 Enfield paper shows quite nicely how AMO can modulate ENSO therefore indicating that pinning much hope on a single indice is bound to disappoint as the chaotic nature of the climate can simply not be represented by something as simple as a temperature anomaly in one part of one ocean.

Enfield, David B., Alberto M. Mestas-Nuñez, and Paul J. Trimble. "The Atlantic multidecadal oscillation and its relation to rainfall and river flows in the continental US." *Geophysical Research Letters* 28.10 (2001): 2077-2080.

Accordingly, I very much like the final conclusion of this paper (see below) and I am certain it is a useful contribution to the scientific community:

'We conclude that while it may seem possible to use historical probabilities to evaluate regions across the globe that are more likely to be at risk of flooding during an El Niño / La Niña, and indeed circle large areas of the globe under one banner of wetter or drier, the reality is much more complex. It is therefore important to undertake research that focusses on the region(s) of interest and consider the impact of ENSO diversity and other drivers of climate variability on the hydrology and flood hazard.'

Trying to link ENSO to local/regional/watershed scale conditions is nothing new and the relatively abundant literature on this topic is ample proof of that. However, this paper definitely provides some originality and novelty by looking at this topic using a much longer time frame and at the global scale. As to the impact at the global scale, the authors were somewhat beaten to this by the recent work of Ward et al. (references 2-3-4). However Ward et al. used the EU-WATCH forcing dataset (ERA40 reanalysis) which cover the 1958-2001 period. To go back further in time, the authors opted to use the ERA-20CM reanalysis. This provides the advantages of a longer time period (which is important for the analysis of rare events) and having access to 10-members to gain additional insights into natural variability of the climatic system and therefore to uncertainty. ERA-20CM is however a more exotic reanalysis since it is only forced with sea-surface temperature and arctic ice data. This downside will be further discussed in the next section. The methodological approach is clearly original and the analysis of the role of ENSO is fair, in opposite to the overhyped approach of modern medias.

The clear differences between precipitation and streamflow analysis noted by the authors is interesting in the context of ENSO but not new and certainly not surprising. For example, numerous climate change impact studies have shown changes in opposite directions for precipitation and streamflows. Hydrologists know very well that streamflows result from a complex non-linear interaction between its two main drivers - precipitation and temperature.

The methodology is solid and I don't find (nor suspect) anything amiss with this paper. I think the methodology is much stronger than in the aforementioned Ward et al. papers.

However, the data presented in this paper is relatively far from its more traditional counterpart using streamflow gauges and weather station data. The experiment performed here is somewhat close to that of a virtual setting and not far removed to what could be done using a GCM for example, and this should be better acknowledged. The ERA-20CM reanalysis has biases (sometimes large) for both temperature and precipitation. There is more uncertainty going back in time in the forcing dataset for ERA-20CM. Global hydrological models are not the same as models typically used and calibrated at the watershed scale.

I fully understand that there are concessions to be made when going to the global and century scales but they could be better acknowledged. I don't demand a validation of all datasets, but this has to be discussed nonetheless. I note for example, that the ENSO years (Nino and Nina) in this paper do not correspond to the ones commonly accepted, even though the general cycles are well reproduced (Figure 1).

I have some concerns with Figure 2 (extended data) with the validity of estimating the 100-year flood with a limited dataset without taking into account the large statistical confidence interval. I realize this was largely an effort to compare themselves against the work of Ward (which use a much shorter dataset...). For validation, I could think of several other possible Figures that would be more useful.

I suggest that the authors further emphasize the fact that the links between a higher probability of mean monthly flows and actual flooding (which is one of the main interest in many places in the world) is not straightforward. While an increased probability of mean monthly flow may be linked to an increased probability of flooding, this is far from a trivial task, and the time and spatial scales used in this study do not allow any firm conclusions to this respect.

Recommendation

I like this paper very much and it is certainly worthy of publication. There is a very large amount of work behind it and the conclusions are useful to the scientific community. I therefore recommend publications with minor revisions to address some of the points made in this review.

Reviewer #3 (Remarks to the Author):

Upon reading the revised manuscript by Emerton et al. "Complex Picture for Likelihood of ENSO-Driven Flood Hazard", submitted to Nature Communications, I find its clarity and content much improved.

Strong points are that the analysis has become transparent and the results more comprehensive through the inclusion of La Niña events in the main text and the analysis of the maximum probability over all months, supplemented by information on the probability for the individual months. The analysis is exhaustive and addresses several weaknesses that were present in the previous draft, including the opacity of the methods and the coarse resolution that was not consistent with the scale of the hydrological processes.

Novel aspects of the manuscript are the use of the ERA-20CM ensemble to derive the river flows over the 20th century, resulting in the ERA-20CM-R reconstruction at 0.5 degrees resolution, and to define the associated uncertainty. As such, the study presents the first global analysis of the influence of ENSO on the probability of discharge extremes over a substantially long period (110 years, comprising 30+33 El Niño and La Niña events) although in terms of analysis and message it is not unlike previous analysis (e.g., studies by Ward, Dettinger etc.). The main strength, therefore, is the extended length of the ensemble of simulated discharge and precipitation from the reanalysis. The validity of the simulated record is assessed against observed discharge and precipitation but the availability of observed data is limited, The reliability analysis does not make use of the full wealth of discharge data.

In addition to the limited validation, one of the weaker aspects of the study is that the hydrology is summarily analysed. While this does not invalidate the main findings of the study, it leaves unanswered which processes explain the difference in sensitivity between precipitation and discharge to ENSO, including any lagged correlation between them. Personally I can imagine situations in which the response of discharge to ENSO is stronger than that of precipitation as stated by Chiew & McMahon but I see this as the exception rather than the rule and probably the result of antecedent conditions. So, the main conclusions are the different sensitivity of precipitation and discharge to ENSO events and the inherent uncertainty, with important implications for our ability (or lack thereof) to predict the influence of ENSO on river flows. The manuscript would gain in strength if it could show more conclusively where ENSO predictions of precipitation have skill to predict stream flow extremes or where additional hydrological information is necessary to do so. While local studies are relevant, global studies have their own merit and this remains underexposed in the current study.

Still, the present findings are potentially of interest to a wider audience.

In terms of the text, the manuscript is in places repetitive and wordy. For example, captions are repeated in the text and the limited usefulness of predictions appears in lines 126 and 211. In line 82 a significant increase is mentioned but the 40% threshold only is introduced in line 157. Similarly, graphs and maps can be improved. Legends can be better structured and the same colour scales used in my opinion for La Niña and El Niño events. Also, La Niña events (how defined)? should be added to Figure 1.

Furthermore, with the additional analysis, the rationale of the study has changed. Not only high flows (not floods per se) but also low flows are considered. Thus, the text should be updated and the title changed accordingly.

All-in-all, the manuscript has improved in quality and gained in depth but some additional revisions are necessary to restructure and straighten the text and figures and possibly strengthen the evidence on which the conclusions are drawn.

Authors' Response to the comments of Reviewer #2 :

Black text - original review

Blue text - second review

Red text - authors' responses to original review

Please note that the changes made to the text in response to Reviewer #2's (with some likely overlap with reviewers #1 and #3) comments during the first round of revisions are highlighted in red in the manuscript file. The changes made during this second round of revisions in response to Reviewer #3's comments are highlighted in purple in the manuscript file.

Review of 'Complex Picture for Likelihood of ENSO-Driven Flood Hazard' by Emerson et al.

I have previously reviewed this paper when it was submitted to [redacted]. It has been redirected to Nature Communications with the option of having the same reviewers assigned. I have not kept a copy of the original submission but as far as I can tell, the paper remains essentially the same. As such, this review is largely based on the original one made earlier this year. I have gone through the paper once again and have updated my review.

Summary of the key results

This paper is a global assessment of the impact of ENSO flooding hazard over the 20th century. The most important result of this paper is that it recognizes that the role of ENSO in flooding is a lot more complex (and I would add subtle) than often reported in the scientific literature. I personally think that the role of ENSO has generally been overstated and, accordingly, I like the general conclusion of this paper.

This paper is a global assessment of the impact of ENSO flooding hazard over the 20th century. The most important result of this paper is that it recognizes that the role of ENSO in flooding is a lot more complex than often reported in the scientific literature, and especially by the media in general. Some of our very recent work over North-America has shown that the role of ENSO on precipitation and discharge has generally been overstated and that other manifestations of natural variability have much more control than ENSO on hydrological conditions. For example, the 2001 Enfield paper shows quite nicely how AMO can modulate ENSO therefore indicating that pinning much hope on a single indice is bound to disappoint as the chaotic nature of the climate can simply not be represented by something as simple as a temperature anomaly in one part of one ocean.

Enfield, David B., Alberto M. Mestas-Nuñez, and Paul J. Trimble. "The Atlantic multidecadal oscillation and its relation to rainfall and river flows in the continental US." *Geophysical Research Letters* 28.10 (2001): 2077-2080.

Accordingly, I very much like the final conclusion of this paper (see below) and I am certain it is a useful contribution to the scientific community

Originality and significance: if not novel, please include reference

Trying to link ENSO to local/regional/watershed scale conditions is nothing new and the relatively abundant literature on this topic is ample proof of that. However, this paper definitely provides some originality and novelty by looking at this topic using a much longer time frame and at the global scale. As to the impact at the global scale, the authors were somewhat beaten to this by the recent work of Ward et al. (references 2-3-4). However Ward et al. used the EU-WATCH forcing dataset (ERA40 reanalysis) which cover the 1958-2001 period. To go back further in time, the authors opted to use the ERA-20CM reanalysis. This provides the advantages of a longer time period (which is important for the analysis of rare events) and having access to 10-members to gain additional insights into natural variability of the climatic system and therefore to uncertainty. ERA-20CM is however a more exotic reanalysis since it is only forced with sea-surface temperature and arctic ice data. This downside will be further discussed in the next section.

Trying to link ENSO to local/regional/watershed scale conditions is nothing new and the relatively abundant literature on this topic is ample proof of that. However, this paper definitely provides some originality and novelty by looking at this topic using a much longer time frame and at the global scale. As to the impact at the global scale, the authors were somewhat beaten to this by the recent work of Ward et al. (references 2-3-4). However Ward et al. used the EU-WATCH forcing dataset (ERA40 reanalysis) which cover the 1958-2001 period. To go back further in time, the authors opted to use the ERA-20CM reanalysis. This provides the advantages of a longer time period (which is important for the analysis of rare events) and having access to 10-members to gain additional insights into natural variability of the climatic system and therefore to uncertainty. ERA-20CM is however a more exotic reanalysis since it is only forced with sea-surface temperature and arctic ice data. This downside will be further discussed in the next section. The methodological approach is clearly original and the analysis of the role of ENSO is fair, in opposite to the overhyped approach of modern medias.

Originality therefore lies in the longer dataset and somewhat more timid conclusions compared to other studied. The clear differences between precipitation and streamflow analysis noted by the authors is interesting in the context of ENSO but not new and certainly not surprising. For example, numerous climate change impact studies have shown changes in opposite directions for precipitation and streamflows. Hydrologist know very well that streamflows result from a complex non-linear interaction between its two main drivers - precipitation and temperature.

We agree with the reviewer that this is well known within the field of hydrology, but unfortunately in the wider community this is often not known or communicated well, and we believe this analysis is an important message for readers outside of the fields of hydrology and meteorology, and we therefore include this in the paper for the wider audience and for context.

Data & methodology: validity of approach, quality of data, quality of presentation

Generally speaking I think the methodology is solid and I don't find (nor suspect) anything amiss with this paper. I think the methodology is much stronger than in the aforementioned Ward et al. papers.

The methodology is solid and I don't find (nor suspect) anything amiss with this paper. I think the methodology is much stronger than in the aforementioned Ward et al. papers.

Nonetheless, the data presented in this paper is relatively far from its more traditional counterpart using streamflow gauges and weather station data.

However, the data presented in this paper is relatively far from its more traditional counterpart using streamflow gauges and weather station data.

Indeed, this paper acts as a counterpoint to the traditional methods, allowing us to conduct a study using spatially distributed data which is not possible through the use of gauge data. We conduct a similar analysis to that produced by IRI for rainfall, which is widely used across various communities.

The experiment performed here is somewhat close to that of a virtual setting and not far removed to what could be done using a GCM for example, and this should be better acknowledged. The ERA-20CM reanalysis has biases (sometimes large) for both temperature and precipitation. There is more uncertainty going back in time in the forcing dataset for ERA-20CM. Global hydrological models are not the same as models typically used and calibrated at the watershed scale.

The experiment performed here is somewhat close to that of a virtual setting and not far removed to what could be done using a GCM for example, and this should be better acknowledged. The ERA-20CM reanalysis has biases (sometimes large) for both temperature and precipitation. There is more uncertainty going back in time in the forcing dataset for ERA-20CM. Global hydrological models are not the same as models typically used and calibrated at the watershed scale.

We agree with the reviewer, and have further discussed the limitations of using the ERA-20CM dataset (lines 255-266).

I fully understand that there are concessions to be made when going to the global and century scales but they could be better acknowledged. I don't demand a validation of all datasets, but this has to be discussed nonetheless. I note for example, that the ENSO years (Nino and Nina) in this paper do not correspond to the ones commonly accepted, even though the general cycles are well reproduced (Figure 1a).

I fully understand that there are concessions to be made when going to the global and century scales but they could be better acknowledged. I don't demand a validation of all datasets, but this has to be discussed nonetheless. I note for example, that the ENSO years (Nino and Nina) in this paper do not correspond to the ones commonly accepted, even though the general cycles are well reproduced (Figure 1).

We again agree with the reviewer, and have addressed this statement in our previous response, through addition of further details of assumptions and limitations. We have also further added discussion of the differences between the ERA-20CM ENSO years and the "commonly accepted" years (lines 277-285). While there are some discrepancies between the two, they generally agree well.

I have some concerns with Figure 2 (extended data) with the validity of estimating the 100-year flood with a limited dataset without taking into account the large statistical confidence interval. I realize this

was largely an effort to compare themselves against the work of Ward (which use a much shorter dataset...). For validation, I could think of several other possible Figures that would be more useful.

I have some concerns with Figure 2 (extended data) with the validity of estimating the 100-year flood with a limited dataset without taking into account the large statistical confidence interval. I realize this was largely an effort to compare themselves against the work of Ward (which use a much shorter dataset...). For validation, I could think of several other possible Figures that would be more useful.

We agree that estimating the 100-year return period using such a dataset has questionable validity, and this is one reason why we have largely based our analysis on percentiles of river flow rather than return periods, and included several further figures aimed at validating the dataset against observations. (Supplementary information Figures 3, 6)

Appropriate use of statistics and treatment of uncertainties

I commend the effort of using all 10 members of the reanalysis. The treatment of uncertainty could be improved by looking at the global distribution of inter-member variability.

We thank the reviewer for their comment and suggestion; we have now extended the analysis of the inter-member variability, and produced and included global maps of the uncertainty across the ensemble members (figures 2b, 4b and 5) in the main text, as suggested. Additional text discussing these figures is included, lines 98-128 and 165-213.

Conclusions: robustness, validity, reliability

I fully agree with the main conclusion of this paper which I have copied below. I find this to be a robust and useful conclusion.

‘We conclude that while it may seem possible to use historical probabilities to evaluate regions across the globe that are more likely to be at risk of flooding during an El Niño / La Niña, and indeed circle large areas of the globe under one banner of wetter or drier, the reality is much more complex. It is therefore important to undertake research that focusses on the region(s) of interest and consider the impact of ENSO diversity and other drivers of climate variability on the hydrology and flood hazard.’

Accordingly, I very much like the final conclusion of this paper (see below) and I am certain it is a useful contribution to the scientific community:

‘We conclude that while it may seem possible to use historical probabilities to evaluate regions across the globe that are more likely to be at risk of flooding during an El Niño / La Niña, and indeed circle large areas of the globe under one banner of wetter or drier, the reality is much more complex. It is therefore important to undertake research that focusses on the region(s) of interest and consider the impact of ENSO diversity and other drivers of climate variability on the hydrology and flood hazard.’

Suggested improvements: experiments, data for possible revision

See earlier comments. I suggest that the authors further emphasize the fact that the links between a higher probability of mean monthly flows and actual flooding (which is one of the main interest in many places in the world) is not straightforward. While an increased probability of mean monthly flow

may be linked to an increased probability of flooding, this is far from a trivial task, and the time and spatial scales used in this study do not allow any firm conclusions to this respect.

I suggest that the authors further emphasize the fact that the links between a higher probability of mean monthly flows and actual flooding (which is one of the main interest in many places in the world) is not straightforward. While an increased probability of mean monthly flow may be linked to an increased probability of flooding, this is far from a trivial task, and the time and spatial scales used in this study do not allow any firm conclusions to this respect.

As this was a complex and extensive analysis, we had to pick a focus point, and as such we look at the historical probabilities of high / low mean monthly river flow, as an indicator of likely increased / decreased flood hazard. There are however many other ways that this could be studied, for example with an analysis of floodiness (Stephens et al., 2016) at the global scale. This would require aggregating the results across regions however, and in this study we decided to focus on producing results at the resolution of the model.

References: appropriate credit to previous work?

All appears OK.

Clarity and context: lucidity of abstract/summary, appropriateness of abstract, introduction and conclusions

All OK.

Recommendation

I like this paper very much and it is certainly worthy of publication. There is a very large amount of work behind it and the conclusions are useful to the scientific community.

I like this paper very much and it is certainly worthy of publication. There is a very large amount of work behind it and the conclusions are useful to the scientific community. I therefore recommend publications with minor revisions to address some of the points made in this review.

We again thank the reviewer for their comments which we believe were helpful in improving this manuscript.

Authors' Response to the comments of Reviewer #3 :

Black text - second review

Purple text - authors' responses to second review

Please note that the changes made to the text in response to Reviewer #2's (with some likely overlap with reviewers #1 and #3) comments during the first round of revisions are highlighted in red in the manuscript file. The changes made during this second round of revisions in response to Reviewer #3's comments are highlighted in purple in the manuscript file.

Upon reading the revised manuscript by Emerton et al. "Complex Picture for Likelihood of ENSO-Driven Flood Hazard", submitted to Nature Communications, I find its clarity and content much improved.

Strong points are that the analysis has become transparent and the results more comprehensive through the inclusion of La Niña events in the main text and the analysis of the maximum probability over all months, supplemented by information on the probability for the individual months. The analysis is exhaustive and addresses several weaknesses that were present in the previous draft, including the opacity of the methods and the coarse resolution that was not consistent with the scale of the hydrological processes.

Novel aspects of the manuscript are the use of the ERA-20CM ensemble to derive the river flows over the 20th century, resulting in the ERA-20CM-R reconstruction at 0.5 degrees resolution, and to define the associated uncertainty. As such, the study presents the first global analysis of the influence of ENSO on the probability of discharge extremes over a substantially long period (110 years, comprising 30+33 El Niño and La Niña events) although in terms of analysis and message it is not unlike previous analysis (e.g., studies by Ward, Dettinger etc.). The main strength, therefore, is the extended length of the ensemble of simulated discharge and precipitation from the reanalysis. The validity of the simulated record is assessed against observed discharge and precipitation but the availability of observed data is limited, The reliability analysis does not make use of the full wealth of discharge data.

We agree that a key strength of this paper lies in the new 20th Century dataset used for the analysis. We would also argue that the message of the paper differs from previous analyses in that we emphasise the uncertainties involved in using ENSO for prediction of flood hazard which are often not communicated well despite being discussed in the literature. Additionally, within the reliability analysis we have included >1000 GRDC stations' discharge data; it is unfortunate for such studies that discharge data availability is severely limited particularly in regions most strongly affected by ENSO.

In addition to the limited validation, one of the weaker aspects of the study is that the hydrology is summarily analysed. While this does not invalidate the main findings of the study, it leaves unanswered which processes explain the difference in sensitivity between precipitation and discharge to ENSO, including any lagged correlation between them. Personally I can imagine situations in which

the response of discharge to ENSO is stronger than that of precipitation as stated by Chiew & McMahon but I see this as the exception rather than the rule and probably the result of antecedent conditions. So, the main conclusions are the different sensitivity of precipitation and discharge to ENSO events and the inherent uncertainty, with important implications for our ability (or lack thereof) to predict the influence of ENSO on river flows. The manuscript would gain in strength if it could show more conclusively where ENSO predictions of precipitation have skill to predict stream flow extremes or where additional hydrological information is necessary to do so. While local studies are relevant, global studies have their own merit and this remains underexposed in the current study.

We certainly agree that it would be interesting to follow-up this paper with further work to look into the processes explaining the differences in sensitivity. This is something that varies across the globe, and this work would merit an entire paper in its own right, and was as such outside of the scope of this paper. Indeed, the results presented here represent a huge amount of analysis, and we also do not wish to over-complicate the message of this particular paper, which is intended for the wider community.

In light of the latter comment, we have amended the figures (2b and 4b) displaying the range across the probabilities (reversing the colour scale) to emphasise where these historical probabilities could potentially be used (i.e. high probabilities and low uncertainty) to predict streamflow extremes. Lines 124-127 were added to reflect these changes.

Still, the present findings are potentially of interest to a wider audience.

In terms of the text, the manuscript is in places repetitive and wordy. For example, captions are repeated in the text and the limited usefulness of predictions appears in lines 126 and 211. In line 82 a significant increase is mentioned but the 40% threshold only is introduced in line 157.

We thank the reviewer for pointing out these oversights. We have re-read the manuscript and removed / amended instances where text was found to be repetitive (e.g. lines 33, 158, 163, 205-210). We have also moved the explanation of the 40% threshold to the first mention of significant influence in line 83. We chose to keep the text regarding uncertainty in both lines 126 and 211 as this repetition is meant to emphasise this key conclusion which is confirmed with the additional analysis presented after line 126, although we have amended the text around line ~211 to make this less repetitive.

Similarly, graphs and maps can be improved. Legends can be better structured and the same colour scales used in my opinion for La Niña and El Niño events. Also, La Niña events (how defined)? should be added to Figure 1.

We agree that the legends could be more informative and better displayed, and as such have amended the legends on figures 2, 3, 4 and 5 in the main text and 3, 4, 5, 7 and 8 in the supplementary information. Changes are to add captions to the legends, and to change the legends of figures 2 and S14 to boxes rather than a colourbar. We decided to keep the different but complementary colour scale for the La Nina maps as it can quickly become confusing and we wish it to be clear which one is being considered at all times. We have added La Nina events to Figure 1, this was indeed an oversight in the previous revision. Additionally, as mentioned, we reversed the colourscale on figures 2b and 4b to give a more clear indication of the regions where the probabilities are more certain.

Furthermore, with the additional analysis, the rationale of the study has changed. Not only high flows (not floods per se) but also low flows are considered. Thus, the text should be updated and the title changed accordingly.

In this study we discuss both abnormally high and low flows, in terms of increased or decreased flood hazard, and as such believe that the title still adequately describes the contents, indeed particularly as we focus throughout the paper on high flows with many of the low flow results included only in the supplementary information. We have clarified in the abstract that we discuss both increased and decreased flood hazard in the paper and do not directly look at floods.

All-in-all, the manuscript has improved in quality and gained in depth but some additional revisions are necessary to restructure and straighten the text and figures and possibly strengthen the evidence on which the conclusions are drawn.